# Nonsense-Mediated mRNA Decay: Mechanisms and Recent Implications in Cardiovascular Diseases

**DOI:** 10.3390/cells14161283

**Published:** 2025-08-19

**Authors:** Fasilat Oluwakemi Hassan, Md Monirul Hoque, Abdul Majid, Joy Olaoluwa Gbadegoye, Amr Raafat, Djamel Lebeche

**Affiliations:** 1Department of Physiology, College of Medicine, The University of Tennessee Health Science Center, Memphis, TN 38163, USA; dhassanf@uthsc.edu (F.O.H.); mhoque5@uthsc.edu (M.M.H.); amajid1@uthsc.edu (A.M.); jgbadeg1@uthsc.edu (J.O.G.); dr.amrraafat@gmail.com (A.R.); 2College of Graduate Health Sciences, The University of Tennessee Health Science Center, Memphis, TN 38163, USA

**Keywords:** nonsense-mediated mRNA decay (NMD), premature termination codon (PTC), up-frameshift proteins (UPFs), cardiovascular diseases (CVDs)

## Abstract

This review highlights the emerging functional implications of nonsense-mediated mRNA decay (NMD) in human diseases, with a focus on its therapeutic potential for cardiovascular disease. NMD, conserved from yeast to humans, is involved in apoptosis, autophagy, cellular differentiation, and gene expression regulation. NMD is a highly conserved surveillance mechanism that degrades mRNAs containing premature termination codons (PTCs) located upstream of the final exon-exon junction. NMD serves to prevent the translation of aberrant mRNA and prevents the formation of defective protein products that could result in diseases. Key players in this pathway include up-frameshift proteins (UPFs), nonsense-mediated mRNA decay associated with p13K-related kinases (SMGs), and eukaryotic release factors (eRFs), among others. Dysregulation of NMD has been linked to numerous pathological conditions such as dilated cardiomyopathy, cancer, viral infections, and various neurodevelopmental and genetic disorders. This review will examine the regulatory mechanisms by which NMD regulation or dysregulation may contribute to disease mitigation or progression and its potential for cardiovascular disease therapy. We will further explore how modulating NMD could prevent the outcomes of mutations underlying genetically induced cardiovascular conditions and its applications in personalized medicine due to its role in gene regulation. While recent advances have provided valuable insights into NMD machinery and its therapeutic potential, further studies are needed to clarify the precise roles of key NMD components in cardiovascular disease prevention and treatment.

## 1. Introduction

Nonsense-mediated decay (NMD) is an evolutionarily conserved pathway present in all eukaryotes, playing a crucial role in regulating mRNA translation. It targets mRNA transcripts containing premature termination codons (PTCs) for rapid degradation, thereby preventing the production of defective proteins that could result in either loss or gain of function [1,2]. Initially identified in *Caenorhabditis elegans* and *Saccharomyces cerevisiae*, NMD has been recognized as a widely conserved quality control process in eukaryotic protein translation [3,4].

Beyond its role in quality control, NMD also regulates gene expression by degrading not only faulty mRNAs but also some normal transcripts [5]. These mechanisms ensure adequate gene expression levels in response to cellular demands [6]. NMD can render nonsense alleles functionally null, thereby influencing the regulation of specific protein levels [4,7]. It is estimated that NMD downregulates approximately one-third of disease-causing mRNA by eliminating aberrant transcripts generated due to routine gene expression errors [6].

This review explores the mechanisms of NMD, its physiological and pathological roles, and its potential as a therapeutic target, particularly in cardiovascular diseases. We also examine how modulating NMD could mitigate the effects of mutations underlying genetically induced cardiovascular conditions.

## 2. Mechanism of NMD and Key Players

NMD is usually triggered when a premature stop codon (UAA, UGA, UAG) is located more than 50–55 nucleotides upstream of the last exon/exon junction in the mRNA transcript [8]. This early occurring stop codon is referred to as a premature termination codon (PTC) [9]. The exon junction complex (EJC) deposited 20–24 nucleotides upstream of exon/exon junctions during splicing, plays a critical role in this process [10]. The existence of EJCs more than 50–55 nucleotides downstream of the PTC defines an NMD substrate [11,12]. However, stop codons located downstream of the final exon junction typically evade NMD.

Nonsense-mediated mRNA decay (NMD) is initiated when a ribosome encounters a PTC upstream of an EJC [13]. This event causes ribosomal stalling and signals the recruitment of a translation termination complex. This complex includes eukaryotic release factors (eRF1 and eRF3) and NMD surveillance factors, notably the ATP-dependent RNA helicase up-frameshift protein 1 (UPF1). UPF1 is then phosphorylated by SMG1 (nonsense-mediated mRNA decay-associated p13K-related kinase). Phosphorylated UPF1 subsequently recruits UPF2, UPF3, and other NMD factors to bridge the EJC and initiate mRNA degradation [6,14]. This entire process constitutes the canonical “EJC model” of NMD (illustrated in Figure 1a). Additional factors, such as SMG5, SMG6, and SMG7, are also recruited by UPF1. These factors, in turn, enlist protein phosphatase 2A to dephosphorylate UPF1 after mRNA degradation, facilitating its recycling [15,16]. Conversely, NMD is not triggered if no PTC is present upstream of an exon/exon junction. In such cases, the ribosome simply reaches the normal stop codon without initiating the decay pathway (as depicted in Figure 1b).

### 2.1. Alternative NMD Mechanisms

An alternative mechanism of NMD activation depends on the distance between PTC and poly (A) site [14]. When this distance is abnormally long, as shown in Figure 1c, premature termination is more likely to occur due to delayed interaction between the terminating ribosome and the poly A binding protein. This is known as the faux 3′UTR model [1]. In this model, UPF1 and PABPC1 compete to bind eukaryotic release factor 3 (eRF3), which, along with eRF1, recognizes termination codons, including PTCs [11,12,17]. If UPF1 binds to eRF first, the mRNA is targeted for degradation; if PABPC1 binds eRF3 first, the mRNA escapes degradation and re-enters a new round of translation [18,19]. This model typically occurs during subsequent rounds of translation, targeting PTC-containing mRNAs that escaped degradation during the pioneer round [18].

### 2.2. Other Unclear Mechanisms of NMD

Despite numerous reports supporting the canonical EJC and 3′Faux models, other alternative mechanisms exist. Some PTC-containing mRNAs evade NMD by reinitiating translation downstream of the early stop codon, as observed in transcripts like the β-immunoglobulin µ heavy chain and BRCA1 [20]. Newer models suggest that UPF1 associates with elongating ribosomes on all translating mRNAs, with UPF2 and UPF3 recruited later along with release factors to regulate premature termination [4]. Some other mechanisms, reported in baker’s yeast for instance, involve the SMG7 orthologue, EBs1, without the other SMGs and UPFs [21,22]. Additionally, an NMD mechanism involving only UPF1 and UPF2, bypassing UPF3 and other canonical components, has been reported in *Trypanosoma brucei* [23].

From these studies, it is evident that newer mechanisms of the NMD continually emerge in various species with novel members of the NMD machinery, and more research is needed to fully understand the other unclear and implicated mechanisms of the NMD pathway. Studies have also suggested that intracellular magnesium (Mg^2+^) levels and its transporter Alrp1 may influence NMD activity [24].

## 3. NMD in Physiological Processes

NMD plays a role in the maintenance of cellular physiology, early organismal development, and responses to various environmental stimuli, such as those that cause apoptosis, maturation, development, and differentiation [6].

### 3.1. NMD and Regulation of Gene Expression

NMD is closely linked to the termination of translation [5]. Recent transcriptome-wide mRNA profiling has identified a vast array of endogenous mRNAs, along with aberrant mRNAs, as NMD targets [25]. Nonsense mutations account for approximately 20% of all disease-associated single-base-pair mutations, many of which introduce PTCs due to errors in nucleic acid metabolism, programmed recombination, or programmed ribosomal frameshifting (PRF) [26,27]. Additionally, NMD regulates alternative splicing, which affects around 12% of natural transcripts and modulates the isoforms of 30% of all expressed genes. This process reduces the NMD substrate by 2 to 20-fold, as alternative splicing has been reported in nearly 95% of all multi-exon genes in mammalian cells [28], which holds significant functional and clinical implications [27,29].

Several genetic disorders, including polycystic kidney disease, muscular dystrophy, and cystic fibrosis, have been associated with PTCs that trigger NMD-mediated degradation of target mRNAs [27]. For example, the NMD pathway involved in the cystic fibrosis transmembrane conductance regulator (CFTR) gene amplifies the effects of nonsense mutations associated with cystic fibrosis. This results in a more severe cystic fibrosis phenotype due to the degradation of RNA and reduced CFTR expression [30]. NMD of β-globin mRNA limits the synthesis of C-terminally truncated dominant negative β-globin chains and thus protects most heterozygotes from symptomatic β-thalassemia [31]. Moreover, alternative splicing contributes to mRNA isoform diversity while simultaneously generating PTCs that designate transcripts for NMD [32].

Although growing evidence supports NMD’s role in regulating gene expression and influencing multiple biological processes, further research is necessary to unravel the full impact of NMD modulation on gene expression.

### 3.2. NMD and Development and Differentiation

NMD plays a dynamic and essential role in developmental and cellular differentiation processes. It targets and degrades approximately one-third of mutated and disease-causing mRNAs, underscoring its pivotal function in maintaining cellular integrity. Knockdown of key NMD factors, including the UPFs and SMGs, has been shown to cause extensive cell death in embryonic tissues, resulting in lethality in zebrafish and mice [33,34,35,36].

Studies further highlight the significance of the NMD factor UPF2 in maintaining hematopoietic stem cells, facilitating T cell maturation, activating resting lymphocytes, supporting fetal liver development, and enabling adult liver regeneration [29,35]. Additionally, NMD activity exhibits variable effects across different stem cell types. For instance, depletion of *UPF3b* inhibits differentiation, whereas depletion of UPF1 prevents proliferation while promoting neuronal stem cell differentiation [37,38]. Furthermore, in developing muscle cells, finely tuned NMD activity plays a crucial role in the differentiation of myoblasts to myotubes [39]. The diverse and context-dependent roles of NMD in development warrant specific attention, as individual NMD factors exert distinct effects that cannot be generalized [40]. A deeper understanding of these mechanisms will be essential for elucidating the precise impact of NMD on cellular differentiation.

### 3.3. NMD and Apoptosis

Accumulation of detrimental proteins activates enzymatic pathways leading to cleavage of proteins and nucleic acids and culminates in cell death in the apoptotic pathway [41]. On the other hand, NMD eliminates mRNAs with PTCs to prevent synthesis of potentially detrimental or non-functional truncated proteins [42]. Studies by Jia et al. showed that caspases, which are used in both intrinsic and extrinsic pathways of apoptosis, cleave two major factors of NMD, UPF1 and UPF2, during the apoptotic processes. This implies that caspase-cleaved UPF fragments upregulate apoptotic stress-related genes, inhibit NMD, and promote cell death [41]. SMG1, another NMD factor, has been associated with the regulation of apoptosis and cellular stress responses, such as DNA damage, and oxidative and hypoxic stresses in *Caenorhabditis elegans* and human [43,44].

NMD promotes timely unfolded protein response (UPR) termination, aiding cell survival under UPR-inducing conditions and reducing the likelihood of stress-induced apoptosis, while NMD-deficient patients exhibit a greater response when subjected to ER stress [45]. However, NMD-deficient (*UPF3b*-null) mice show elevated liver apoptosis in response to external ER stress [45]. Therefore, the interplay between ER stress, apoptosis, and NMD harnesses the potential for developing strategies to treat diseases characterized by chronic ER stress [45,46]. The combined effect of NMD and upregulated pro-apoptotic genes has been closely linked to cell death in cancer cells. Such is the role of GAS5 (growth arrest-specific 5) RNA, which binds to and antagonizes glucocorticoid receptors [47,48].

### 3.4. NMD and Autophagy

Autophagy is a lysosome-dependent catabolic process where cytoplasmic components are sequestered in double-membrane autophagosomes and subsequently degraded following fusion with lysosomes [49]. This serves as a cellular surveillance mechanism to eliminate detrimental misfolded/aggregated cytoplasmic proteins and provides a source of amino acids during metabolic stress [49]. While autophagy is a physiological process, NMD rapidly degrades the selected mRNAs and contributes to DNA damage response, cell cycle regulation, cell viability, and viral infections [50].

Inhibition of the NMD by different cellular stresses, such as hypoxia, amino acid deprivation, viral infection, or other environmental insults, overloads the ER with misfolded, mutated, and aggregated proteins [51]. These stressful conditions could further generate stress signals and stimulate autophagy [52]. The inhibition of NMD induces autophagy both in vitro and in vivo, whereas hyperactivation of NMD blunts its induction under various cellular stressors [51]. In addition, depletion of activating transcription factor 4 (ATF4), a transcription factor for autophagy genes, and UPF2 (an NMD factor), results in decreased autophagy compared to depletion of only UPF [51]. This indicates that NMD shapes the autophagy pathway and both pathways are intertwined [51].

## 4. NMD Escape Mechanism

The efficiency of NMD varies across transcripts, cells, tissues, and individuals, allowing some mRNA to evade NMD surveillance even in the presence of NMD-inducing factors [11,53]. This ability of some PTC-containing mRNA to evade NMD is termed NMD escape. This variability affects the phenotypic manifestation of disease-causing mutations and plays a role in the clinical interpretation of genetic disorders. NMD escape may occur when the ribosome fails to recognize the PTC and reads through it or due to differences in cellular NMD efficiency [11]. As a result, some cells rapidly degrade nearly all PTC-containing mRNAs, while others permit these mRNAs to bypass NMD almost entirely [11].

### 4.1. NMD Escape in Tumor Cells

NMD plays a crucial role in quality control and post-transcriptional gene expression regulation across various cellular processes [54]. Depending on the specific model, it can function as both a tumor suppressor and a pro-tumorigenic factor in cancer progression [55]. However, tumor cells utilize diverse regulatory mechanisms to modulate NMD efficiency, promote escape, and generate potentially immunogenic truncated proteins. These mechanisms include reducing translational read-through, inducing cellular stress conditions, inactivating translation, altering NMD efficiency, and inhibiting post-translational processes [11].

NMD escape has been studied in cancer genes using the Cancer Genome Atlas dataset [56]. Twenty-nine genes with altered NMD rates have been described, revealing novel potential tumorigenic genes and pinpointing candidate driver mutations in established cancer genes, thus enabling further functional characterization [57]. A study focusing on NMD in colorectal cancers (CRCs), particularly those with microsatellite instability (MSI), demonstrated a connection between high PTC mRNA levels in MSI CRC and increased expression of UPF1/2 and SMG1/6/7 [58]. However, inhibiting or bypassing NMD restores the expression of PTC-mutant proteins, such as HSP110DE9, which plays a beneficial role in suppressing MSI CRC tumor growth. Assessing NMD activity in cancer cells provides valuable clinical insights into tumors with high NMD activity, and this may improve the treatment of primary tumors [59].

### 4.2. NMD Escape and Its Role in Immunity

Disrupting NMD, as demonstrated by SMG7 knockout in a fibrosarcoma cell line, suppresses oncogenic properties by downregulating matrix metalloprotease 9 (MMP9) [55]. This process suggests a potential link between NMD inhibition and modulation of neoantigen production, highlighting NMD as a potential therapeutic target in cancer treatment. These neoantigens, resulting from mutations, can be recognized by the immune system, triggering an immune response and enhancing anti-tumor immunogenicity [56]. Genome-wide screens have identified factors involved in NMD-linked protein quality control, separate from the canonical ribosome/quality control pathway [60]. This pathway emphasizes NMD’s role in downregulating key tumor suppressor genes, providing a valuable framework for characterizing relevant factors in cancer progression.

The success of immunotherapy can be affected by NMD. A study found that frameshifted transcripts escaping NMD serve as strong predictors of positive responses to cancer immunotherapy. This finding suggests that targeting NMD escape mechanisms may enhance the effectiveness of cancer immunotherapy [61]. Other studies have further demonstrated that patients harboring NMD escape mutations, particularly frameshift indels (fs-indels), show a higher likelihood of clinical benefit in checkpoint inhibitor therapy [56]. Hence, the rate of NMD escape mutations may serve as a significant biomarker linked to therapeutic response. Even a single NMD escape mutation could produce a substantial immunogenic effect, reinforcing the potential of these mutations as key indicators for immunotherapy success [56].

## 5. Inhibition of Nonsense-Mediated mRNA Decay

Several genetic mutations linked to human genetic diseases lead to the production of transcripts containing PTCs, resulting in their degradation via NMD. This biological mechanism likely serves to eliminate non-functional proteins at the transcript level, preventing potential dominant-negative effects on cellular processes [62]. However, one drawback of this system is the risk of haploinsufficiency. In cases where the mutant protein does not significantly impair transcript function, the selective inhibition of NMD may offer a viable strategy to rescue the associated phenotype [63]. Inhibition of NMD may provide therapeutic success for some NMD-associated diseases.

Certain microRNAs have been reported to inhibit NMD in various organs. For instance, miR-128 plays a role in regulating the NMD circuit in neurons, thereby influencing neural fate and function across multiple regions of the central nervous system [64,65,66]. Additionally, miR-4651 regulates NMD by targeting SMG9 mRNA, which is involved in PTC recognition [67], while miR-433 negatively modulates NMD by targeting SMG5 mRNA, a crucial factor in degrading targeted mRNAs [68]. NMD activity can also be impaired by apoptotic pathways proteases, known as caspases. Caspase 3 and 7 have been shown to cleave both UPF1 and UPF2, key factors necessary for initiating mRNA degradation [41].

Several compounds have been reported to influence NMD efficiency. For example, 5-azacytidine, previously used to treat myeloid leukemia, has an inhibitory effect on NMD [69]. Curcumin has also been found to reduce the expression of NMD factors UPF1, UPF2, and UPF3 [70]. Additionally, wortmannin, an experimental NMD inhibitor, has been shown to block SMG-1, a PI3-kinase-related protein responsible for phosphorylating UPF1 during the NMD process. While wortmannin could potentially be used to modulate the NMD pathway and mitigate the effects of certain genetic mutations [62], its harmful cellular impact makes it unsuitable for human medical treatment [71]. Furthermore, ataluren (PTC124), a compound that facilitates ribosomal read-through, has been reported to enhance BMPR2 protein expression in cells derived from pulmonary arterial hypertension patients with nonsense mutations [72]. These small compounds could aid in gene therapy in patients affected by these specific diseases/mutations.

Another notable regulator of NMD is Mammalian Staufen1 (STAU1), which plays a role in STAU1-mediated mRNA decay (SMD) [66]. SMD competes with NMD for UPF1, which is a common factor required for their activity [39]. STAU-1 binds to UPF1 and initiates RNA decay [73]. This competitive interaction between SMD and NMD to UPF1 results in reciprocal inhibition, where one pathway’s activity restricts the other. Studies have also shown that downregulation of another NMD factor, UPF2, inhibits NMD while increasing SMD activity due to the overlapping binding site of UPF2 and STAU1 on UPF1 [39]. Another study has also shown that during adipogenesis, NMD efficiency decreases due to the interference by RNA-mediated depletion of NMD components, which are upregulated during adipogenesis [74]. Consequently, an increase in the activity of SMD is observed.

## 6. Nonsense-Mediated mRNA Decay in Cardiovascular Disease Conditions

While NMD serves as a protective process, its dysregulation has been reported to disrupt cellular homeostasis, contributing to the pathogenesis of various disorders [75], including cardiovascular diseases [76], developmental anomalies [77], neurological disorders [78], cancer [79], genetic syndromes [7], and immune response [80], as depicted in Figure 2. In some cases, excessive NMD activity can lead to the degradation of essential transcripts, exacerbating the underlying disease conditions.

In the context of cardiovascular diseases, NMD has been implicated in conditions such as dilated cardiomyopathy [81], hypertrophic cardiomyopathy [82], and hypertension [83], where mutations in genes critical for heart muscle function trigger this surveillance mechanism, leading to the degradation of essential proteins required for cardiac function. While NMD serves as a protective mechanism against potentially harmful truncated proteins, its excessive activation can contribute to disease progression by reducing the expression of necessary cardiac proteins. Hence, understanding the balance between NMD’s protective and pathological roles could pave the way for novel treatments targeting heart disease at the molecular level.

### 6.1. Cardiac Conduction Disease

The phenotypic spectrum of TNNI3K mutations, a dual-function kinase with four domains, has been recognized as a significant factor in cardiovascular disease due to its association with heart failure and hypertrophy [84]. This highlights the relevance of NMD in understanding genotype/phenotype correlations in cardiac conduction disease (CCD) [85]. NMD dysfunction leads to the accumulation of truncated proteins, which can have deleterious effects on cardiac function [7]. Additionally, NMD has been found to regulate the expression of genes involved in cardiac development and function [86,87]. In congenital long QT syndrome type 2 (LQT2), the frameshift mutation P926AfsX14 undergoes NMD, leading to reduced mRNA, protein, and hERG current expression. This degradation contributes to severe clinical phenotypes, especially when combined with environmental triggers and genetic modifiers, emphasizing the pathogenic role of NMD in LQT2 [88]. Furthermore, the Q1070X nonsense mutation in LQT2 has demonstrated that inhibiting NMD can restore functional expression of mutant channels, suggesting that targeting NMD processes could serve as a potential therapeutic approach for patients carrying nonsense and frameshift mutations in LQT2 [89].

### 6.2. Dilated Cardiomyopathy

In dilated cardiomyopathy (DCM), a missense mutation in the LMNA gene disrupts a normal splicing site, triggering NMD and leading to a significant reduction in lamin A and C expressions [81]. This mutation has been linked to laminopathies, which manifest as DCM, atrial, and ventricular arrhythmia [81,90]. NMD has also been shown to significantly downregulate the expression of mutant transcript in dilated cardiomyopathy caused by the missense LMNA gene mutation [81,90,91]. A reduction in the efficiency of NMD processes has also been reported to be associated with titin-truncating variants (TTNtvs), which are the most common genetic cause of dilated cardiomyopathy [92]. A mutation in titin (TTN), which is a large sarcomeric protein, causes an accumulation of truncated titin proteins that impairs cardiac contractile ability [93]. Consequently, cytoskeletal proteins encoded by TTN are reduced, and this results in a reduced force of transmission within the heart, leading to DCM [94]. Another study has reported that reduction in TTN and lamin A/C triggered by NMD can reduce cardiac metabolism of medium and long chain fatty acids, thereby increasing cardiac glycolysis dependency which could consequently increase activation of mammalian target of rapamycin target (mTORC1) signaling pathway [95], hence promoting ineffective protein synthesis and autophagy leading to myocardial damage, myocardial contraction disorder, and formation of DCM [96]. NMD has been proposed as a co-existing mechanism that degrades the mutated allele and reduces lamin expression, which has been implicated in the suppression of healthy alleles, ultimately contributing to DCM progression [90].

### 6.3. Hypertrophic Cardiomyopathy

A study conducted in a mouse model of hypertrophic cardiomyopathy (HCM) found that a single mutation in the MYBPC3 gene, encoding the myosin-binding protein C (cMyBP-C) protein, resulted in low cMyBP-C mRNA expression. Surprisingly, three mutants of c-MyBP-C exhibited lower expression levels than control animals [87]. It was demonstrated that NMD specifically influences the abundance of mRNA containing PTCs associated with the cMyBP-C gene, rather than affecting other cMyBP-C transcripts by decreasing the expression of a mutant of the gene. However, NMD inhibition by cycloheximide and emetine significantly increased the mutant expression [87]. Another study showed that cMyBP-C is required to reduce actin/myosin interaction at low calcium concentrations, allowing for complete relaxation during diastole [97]. However, a mutation of cMyBP-C in mice resulted in altered cardiac relaxation at diastole, leading to HCM development [2]. Another study also demonstrated that mutations in cMYBP-C and MYBPC3 increase PTCs and reduce functional cMYBP-C, contributing to the progression of HCM. Additionally, this study highlights the involvement of UPF3 in the mRNA degradation process during translation at the Z disc [82].

In familial hypertrophic cardiomyopathy (HCM) in humans, frameshift mutations in the MYBPC gene leading to an internal truncation of the transcript were found to result in a reduced occurrence of PTCs in the mRNA compared to control mRNA [98]. This suggests NMD might play a role in this process, although further investigation is required to understand the mechanisms involved.

## 7. Potentials of NMD in the Prevention and Treatment of Cardiovascular Disease

There is growing interest in how NMD modulation might be leveraged in cardiovascular diseases, as well as other diseases such as genetic diseases, neurological disorders, and cancer. While in vitro studies show promising effects of NMD inhibition in cardioprotective protein expression [87], the connection between NMD dysregulation and cardiovascular diseases remains less well established. It is plausible that aberrant NMD activity may affect the expression of mRNAs critical for cardiac function. Changes in mRNA stability and processing can profoundly influence protein levels in cardiovascular tissues. Hence, exploring how targeted modulation of NMD through fine-tuning NMD may correct pathogenic mRNA imbalances or reduce the burden of aberrant proteins, and restoring normal function may pave the way for therapeutic innovations. Several small molecules, gene therapies, and other interventions that can modulate NMD activity are currently being investigated [99]. Therefore, targeted manipulation of NMD holds the potential to mitigate disease progression with the promise of reducing the expression of genes associated with cardiovascular disease pathology while preserving the integrity of essential genes [100,101]. Nevertheless, this area remains speculative and requires additional focused research to firmly establish NMD therapeutic strategies for heart disease.

Nonsense-mediated decay acts as a double-edged sword, which could prevent diseases by degrading mRNA with PTC and could also be detrimental if overexpressed [102]. Due to the dual effect of NMD in disease prevention or progression, the challenge of determining the effect of overexpressing NMD factors in CVD prevention must be carefully considered. Interestingly, UPFs, which are key NMD factors, have been reported to be involved in cardiovascular diseases [103,104]. Genetic manipulation of NMD factors might be a potential therapeutic approach to preventing NMD in cardiovascular diseases caused by excessive destruction of essential genes with PTC, such as in congenital long QT syndrome type 2 (LQT2), leading to cardiac conduction disease [105]. This therapeutic approach could also be explored in dilated cardiomyopathy caused by TTN deficiency as a result of NMD activation, which degrades titin proteins. Moreover, research on the Q1070X nonsense mutation in LQT2 has shown that suppressing NMD can restore the proper function of mutant channels. This finding suggests that modulating NMD mechanisms could be a promising therapeutic strategy for patients with nonsense or frameshift mutations in LQT2.

Several small molecules are under investigation for their ability to modulate or inhibit the NMD pathway. For instance, a hydrophobic tetracyclic indole derivative named NMD inhibitor 1 (NMDI 1) has been identified as a specific inhibitor of NMD downstream of UPF3 or UPF2 recruitment and upstream of UPF1 functions [106,107]. This inhibitor does not affect the interaction between UPF1 and UPF3X and would not prevent recruitment of hUPF1 to the EJC; rather, it stabilizes the hyperphosphorylated form of UPF1 [108]. Such molecules have mostly been examined in the context of neurodegenerative or genetic disease models; these molecules are now being repurposed to explore their cardioprotective effects in preclinical studies. For instance, in inherited cardiomyopathies caused by nonsense mutations, a hyperactive NMD system may exacerbate protein insufficiency in the heart. By using small-molecule inhibitors to reduce NMD activity selectively in the myocardium, researchers hope to increase the expression of functional proteins, partially compensating for the underlying mutation. Additionally, given that cardiac stress (such as ischemia or pressure overload) can alter NMD dynamics, transient modulation of the pathway could help attenuate maladaptive remodeling or apoptosis.

NMDI-14 has been reported to increase NMD processes in multiple cell lines [109] and, therefore, increase the expression of endogenous NMD targets in cells [110]. This inhibitor has been shown to restore mRNA integrity despite the presence of PTC, and this could be used as a therapeutic strategy to restore protein expression in NMD-induced CVD despite the presence of missense mutations in the gene, such as in LMNA gene mutation resulting in DCM, or MYBP-C mutation implicated in hypertrophic cardiomyopathy. High-throughput screening combined with structure-based drug design is underway to discover compounds that not only inhibit NMD effectively but also offer favorable pharmacokinetic profiles and tissue specificity—essential properties for cardiovascular applications. Future research is expected to focus on refining these inhibitors for selectivity, optimizing dosing regimens, and elucidating the detailed molecular consequences of NMD modulation in the heart.

## 8. Cross-Disease Insights into NMD: Implications for Cardiovascular Pathophysiology

While NMD serves as a protective process, its dysregulation has been reported to disrupt cellular homeostasis, contributing to the pathogenesis of various disorders [75], including cardiovascular diseases [76], developmental anomalies [77], neurological disorders [78], cancer [79], genetic syndromes [7], and immune response [80]. In some cases, excessive NMD activity can lead to the degradation of essential transcripts, exacerbating the underlying disease conditions. Conversely, certain cancers exploit NMD to degrade tumor-suppressor mRNAs, promoting malignancy [81] (Figure 2).

Insights from neurodevelopmental, virological, oncological, and genetic disease models suggest that the heart, a highly energy-demanding organ with stringent transcriptomic control, may similarly rely on NMD to maintain cellular integrity, especially under pathological conditions. This positions NMD as a promising yet underexplored therapeutic target in cardiovascular diseases (CVDs).

In neurodevelopment, NMD governs key processes such as synaptic plasticity, neuronal differentiation, and learning through fine-tuned transcriptome regulation [111,112]. Its disruption, via mutations in UPF2, UPF3A, or UPF3B, leads to inflammation, altered gene expression, and homeostatic imbalance in disorders such as autism, schizophrenia, and intellectual disability [113,114,115]. Similar transcriptomic sensitivity exists in cardiomyocytes, particularly under stress conditions such as hypoxia or hypertrophy, suggesting that maladaptive NMD activity may exacerbate inflammatory or fibrotic signaling. For instance, UPF2-associated inflammation in the brain [114] may reflect comparable pathways in vascular tissues that influence endothelial function or myocardial remodeling.

Emerging evidence indicates that the loss of certain proteins, including the fragile X mental retardation protein (FMRP), can disrupt the regulation of the nonsense-mediated mRNA decay (NMD) machinery in multiple human cell types. In particular, FMRP deficiency has been shown to lead to an upregulation of the core NMD factor UPF1, which in turn causes hyperactivation of the NMD pathway [116]. Similarly, alterations in NMD regulation with features of hyperactivation have been observed in fibroblasts derived from patients with amyotrophic lateral sclerosis (ALS) [117]. These findings suggest that modulating NMD efficiency, either by reducing or overexpressing key components of the pathway, might represent an attractive therapeutic strategy in neurodegenerative conditions, and this might also be explored to determine if such proteins could improve cardiovascular disease outcomes.

In cancer, NMD displays a dual role, suppressing tumors by degrading oncogenic transcripts or promoting tumor survival by removing truncated tumor-suppressor mRNAs [47,100]. Inhibition of NMD can restore expression of functional truncated p53 isoforms, reactivating apoptotic pathways [118]. This concept is highly relevant to cardiac biology, where p53, ER stress, and apoptosis also dictate cardiomyocyte fate in ischemia or heart failure [119]. For example, NMD suppression increases resistance to ER stress in cancer [48], raising the possibility that modulating NMD in the heart may enhance cell survival during reperfusion injury or metabolic dysfunction. Additionally, NMD factors such as UPF1 and SMG9 influence tumor proliferation and cell migration [68,120], suggesting their possible roles in cardiac fibrosis or post-injury remodeling.

In genetic disorders, the context-specific impact of NMD is evident. While it protects against dominant-negative effects in recessive mutations, NMD can also exacerbate disease by eliminating mRNAs that would otherwise translate into partially functional proteins [7]. Variability in NMD efficiency among individuals [11] implies a role in phenotypic diversity of inherited cardiomyopathies and channelopathies. In diseases like cystic fibrosis, Duchenne muscular dystrophy (DMD), and spinal muscular atrophy, therapies that combine NMD inhibition with read-through agents or CRISPR-based editing have restored functional protein expression [30,121,122]. These strategies may be directly applicable to cardiovascular disorders arising from nonsense mutations in cardiac structural proteins or ion channels, where restoring even partial protein function could improve clinical outcomes. Notably, reduced dystrophin expression in DMD was linked to transcriptional limitation rather than NMD activity [110], prompting caution and reinforcing the need for context-specific investigations of transcriptional versus post-transcriptional regulation.

Altogether, these interdisciplinary findings converge to suggest that NMD is a critical determinant of cardiovascular health, influencing gene expression, immune signaling, stress responses, and possibly therapeutic response. Future CVD research should focus on mapping NMD-regulated transcripts in the heart under various conditions (e.g., hypoxia, viral infection, ER stress), identifying cardiac-specific NMD modulators, and exploring personalized strategies based on individual NMD efficiency. Therapeutically, modulating NMD may offer new avenues to restore defective protein expression, control inflammation, and protect cardiomyocytes.

## 9. Conclusions and Future Perspectives

NMD plays a crucial role in cellular homeostasis by targeting mRNAs with PTCs. Its broad regulatory influence extends to approximately 10% of human cell mRNAs, regulating essential physiological processes and contributing to various diseases. Given its ability to regulate gene expression, NMD presents promising therapeutic opportunities as its effective regulation could drive the progress or prevent diseases. However, despite its widespread impact across physiological and pathological conditions, further research is warranted to overcome key challenges, including specific mutation effects, different individual responses, translational effects in clinical trials, stability of compounds, and off-target effects.

The possibility of NMD being used as a clinical biomarker for CVD is an interesting approach that is worth considering. NMD activity has been reported to determine the extent of tumor growth and predict the effectiveness of NMD-based therapies [123,124]. Nanopore sequencing could identify NMD target genes and measure NMD activity, and this could be considered as a tool to quantify NMD activity to determine the prognosis of diseases [125].

The identification of NMD escape mechanisms holds promise for cardiovascular disease therapy, as PTC-containing genes or genes with missense mutations could escape degradation and proceed to produce the protein needed to prevent these diseases. However, more research is necessary to determine the exact point at which NMD should be inhibited or allowed to maintain a balance. Advancing our understanding in these areas will be essential for optimizing NMD-based approaches for diagnostic and therapeutic applications.

It is also important to determine if NMD could be triggered pharmacologically to target mRNA that escaped NMD. This could be an approach to prevent diseases caused by the presence of aberrant proteins that escaped NMD, as seen in the β-globin gene in β thalassemia. Overall, more research is needed to determine the exact point at which an activation or inhibition of NMD will be most beneficial to prevent diseases. Because NMD serves as an essential quality-control mechanism, global inhibition could lead to the accumulation of aberrant mRNAs and potentially toxic protein products. Therefore, specificity, both in terms of the spectrum of mRNAs affected and tissue targeting, is vital.

Looking ahead, the future of NMD research in cardiovascular diseases appears promising, with continuous advancements in genomic technologies and the refinement of NMD-targeted therapies. Despite the remaining challenges, the dynamic progress in NMD-related therapy for cardiovascular research offers optimism for improved prevention, diagnosis, and treatment, ultimately contributing to better clinical outcomes for patients with cardiovascular diseases. Integrating NMD inhibitors with other modalities, such as gene therapy approaches involving CRISPR/Cas9-mediated gene correction, antisense oligonucleotides that bind to specific RNA regions, or read-through compounds (e.g., ataluren), may provide synergistic benefits targeting the complex molecular landscape of cardiovascular disorders.

## Figures and Tables

**Figure 1 cells-14-01283-f001:**
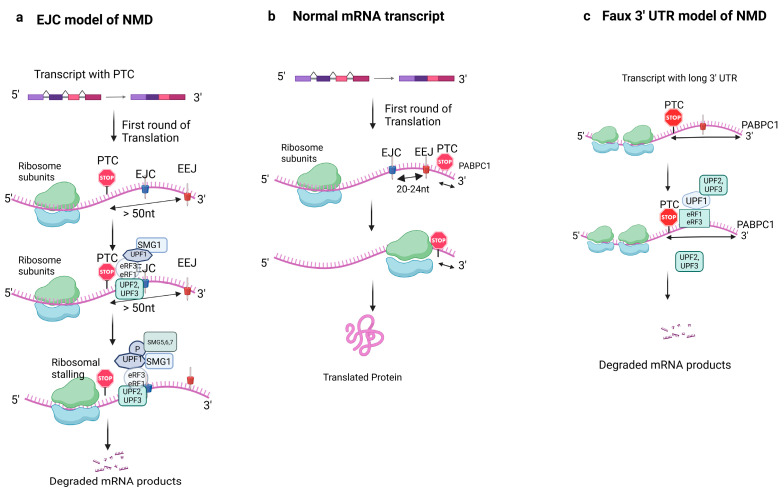
Overview of the NMD pathway (EJC model, Faux 3′UTR model) in eukaryotes. During the first round of translation, the ribosome removes the exon/exon junction complexes and reads the mRNA transcript until it reaches the stop codon. In a healthy mRNA transcript (**b**), the termination codon is downstream within ~50 nucleotides of the EJC, hence translation proceeds. However, if the termination codon is upstream of EJC, the mRNA transcript becomes degraded by NMD. A premature stop codon (UAA) greater than 50 nucleotides upstream of EEJ is recognized and leads to recruitment of eRF1 and eRF3, UPF1, and SMG1, which leads to the formation of the SURF complex; UPF1 is phosphorylated by SMG1, the phosphorylation of UPF1 recruits UPF2, UPF3, SMG5, SMG6, and SMG7 and interacts with other factors such as PNRC2, which leads to the degradation of the mRNA products, as shown in (**a**). The 3′Faux model of NMD is depicted in (**c**), where the distance between the premature termination codon and poly (A) tail contributes to the regulation of the translation. In a normal mRNA transcript, the termination takes place close to the poly (A) site, which enables efficient interaction between the PABP and eRF3 and terminates ribosomal translation (**b**). However, if the stop codon is far upstream of the poly A site, the terminating ribosome cannot interact with PABP efficiently and instead interacts with UPF factors. UPF1 binding to eRF leads to degradation of the mRNA (**c**). Abbreviations: EJC: exon junction complex; EEJ: exon/exon junction; UAA: stop codon; PTC: premature termination codon; UPF: up-frameshift protein; SMG: a serine/threonine-protein kinase; PABPC1: poly A binding protein cytoplasmic 1. (Image created with BioRender).

**Figure 2 cells-14-01283-f002:**
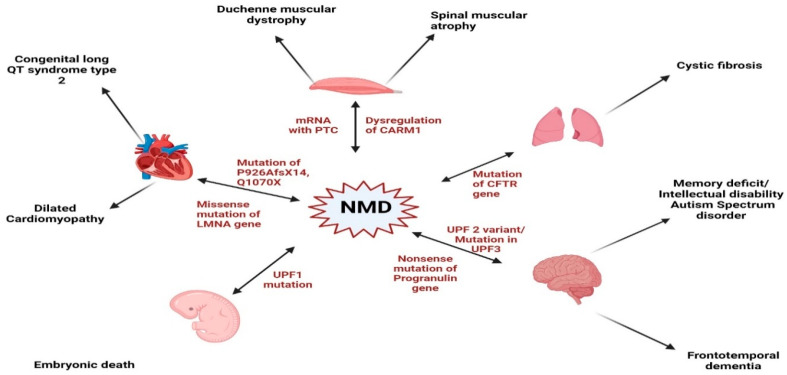
NMD dysregulation: Dysregulation of nonsense-mediated mRNA decay has been linked to different pathological conditions, including cardiovascular diseases, neurological problems, cancer development, genetic disorders, different developmental anomalies, and immune dysregulation. Unraveling the intricate role of NMD in a wide array of diseases fosters the potential of novel therapeutics and interventions in the pathological mechanisms of those diseases. Abbreviations: CARM1: Coactivator-Associated Methyltransferase-1; CFTR: Cystic Fibrosis Transmembrane Conductance Regulator; UPF: Up-Frameshift Protein; PTC: Premature Termination Codon. (Image created with Biorender).

## Data Availability

No new data were created or analyzed in this study.

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
