# Peer review of "Nonsense-Mediated mRNA Decay: Mechanisms and Recent Implications in Cardiovascular Diseases"

_cells, 2025, doi:10.3390/cells14161283_

Round 1

Reviewer 1 Report

Comments and Suggestions for Authors

The authors present a review article discussing the molecular mechanisms of Nonsense-Mediated mRNA Decay (NMD) and its recent implications in cardiovascular and other diseases. While the review attempts to cover a broad range of topics, it contains significant inaccuracies in the description of NMD mechanisms and includes figures with multiple misleading or incorrect elements. In addition, the manuscript lacks clear organization, contains redundant content, and fails to establish a focused narrative. A major revision, including full structural reorganization, is necessary. At this stage, detailed line-by-line feedback is not feasible; instead, I provide the following major concerns and suggestions for improvement:

  1. Title does not reflect the content
    Although the title emphasizes cardiovascular diseases ("Nonsense-Mediated mRNA Decay: Mechanisms and Recent Implications in Cardiovascular Diseases"), the article also covers a wide range of unrelated topics, including cancer, neurodevelopmental disorders, and viral infection. The authors should revise the title to better reflect the actual scope and emphasis of the review.
  2. Major issues in the NMD mechanism section
    The section on molecular mechanisms of NMD requires substantial revision. A clear, logical sequence should be followed:
  • Start with premature termination codon (PTC) recognition by eukaryotic release factors (eRFs)
  • Then describe the stepwise recruitment of NMD factors (e.g., UPF1, UPF2, UPF3, SMG proteins)
  • Conclude with downstream events such as endonucleolytic and exonucleolytic mRNA degradation

Currently, the narrative lacks coherence and contains several inaccuracies. For example, the figure incorrectly suggests that ribosomes do not recognize PTCs and misrepresents the role of eRFs. Furthermore, the discussion of the exon junction complex (EJC) is repetitive and not well integrated into the mechanism.

Figure 1 is especially problematic: it depicts mRNA without EJCs after the first round of translation, followed by a panel showing EJCs reappearing without explanation. This is misleading. The figure and associated text should be revised to reflect the correct sequence of molecular events. The authors should eliminate redundancy, clearly define the role of each component, and consider simplifying this section. Given that many detailed reviews on NMD mechanisms already exist, the authors may consider condensing the mechanistic discussion and placing greater emphasis on disease relevance, which would better serve the review's value.

  1. Clarify models with a unified figure
    The EJC-dependent model and the 3'-UTR (faux 3') model should be illustrated together in a single schematic figure with brief, clear explanations of each. This would reduce redundancy and enhance clarity. Additionally, it is misleading that only the EJC-dependent model is mentioned in the abstract, which does not reflect the broader discussion in the text.
  2. Redundant sections on cancer
    There is overlap between Section 4.1 (“NMD Escape in Tumor Cells”) and Section 6.4 (“NMD and Cancer”), resulting in redundancy. These sections should be merged or their individual purposes clearly distinguished.
  3. Repetitive content in Section 3: NMD in Physiological Processes
    This section also requires major revision. Several parts are repetitive—for example, Section 3.1 discusses genetic diseases under “NMD and Regulation of Gene Expression,” while similar content appears again in Section 6.5 (“NMD and Genetic Diseases”). The authors should avoid overlap and reorganize for clarity.
  4. Unclear distinction between NMD escape and inhibition
    The terms “NMD escape” and “NMD inhibition” are used but not clearly defined. The authors should explain the difference explicitly, ideally with representative examples, and ensure consistent usage throughout the manuscript.

Summary
In summary, the manuscript needs intensive revision for both scientific accuracy and structural clarity. Simplifying the mechanistic details, removing redundancy, and focusing the narrative around clearly defined sections—especially on disease relevance—would substantially improve the quality of the review.

Author Response

We deeply appreciate your comprehensive review of our manuscript and the invaluable comments and feedback you have provided. We have thoroughly considered each of your suggestions and made the necessary revisions to address the raised issues. As suggested, we have removed unnecessary duplications and have improved structures to enhance readability and precision. These adjustments aim to make the manuscript more accessible and coherent for the readers. Based on the revisions made, we believe that the concerns you raised have been adequately addressed.

Following are our responses to the concerns raised.

Comment 1: Title does not reflect the content
Although the title emphasizes cardiovascular diseases ("Nonsense-Mediated mRNA Decay: Mechanisms and Recent Implications in Cardiovascular Diseases"), the article also covers a wide range of unrelated topics, including cancer, neurodevelopmental disorders, and viral infection. The authors should revise the title to better reflect the actual scope and emphasis of the review.

Response 1: The other unrelated disease condition has been removed, and the review is more focused on cardiovascular diseases

Comment 2: Major issues in the NMD mechanism section
The section on molecular mechanisms of NMD requires substantial revision. A clear, logical sequence should be followed:

Response 2: Major revision has been performed on the mechanism and correction of figure 1 has been made. Also figure 1 and 2 has been unified/merged as suggested. 

Comment 3: Clarify models with a unified figure
The EJC-dependent model and the 3'-UTR (faux 3') model should be illustrated together in a single schematic figure with brief, clear explanations of each. This would reduce redundancy and enhance clarity.

Response 3: Both models has been merged into a single figure with a more concise explanation

Comment 4: Redundant sections on cancer
There is overlap between Section 4.1 (“NMD Escape in Tumor Cells”) and Section 6.4 (“NMD and Cancer”), resulting in redundancy. These sections should be merged or their individual purposes clearly distinguished.

Response 4: Section 6.4 has been removed and important content merged with other section

Comment 5: Repetitive content in Section 3: NMD in Physiological Processes
This section also requires major revision. Several parts are repetitive—for example, Section 3.1 discusses genetic diseases under “NMD and Regulation of Gene Expression,” while similar content appears again in Section 6.5 (“NMD and Genetic Diseases”). The authors should avoid overlap and reorganize for clarityResponse 

Response 5: Repetitive content has been removed, and section 6.5 on NMD and genetic diseases has been deleted to avoid redundancy and a non-specific review focus.

Comment 6 : Unclear distinction between NMD escape and inhibition
The terms “NMD escape” and “NMD inhibition” are used but not clearly defined. The authors should explain the difference explicitly, ideally with representative examples, and ensure consistent usage throughout the manuscript.

Response 6: This has been addressed and both terms are clearly defined.

Reviewer 2 Report

Comments and Suggestions for Authors

Comments

  1. In abstract, authors should slightly expand the abstract to clearly explain NMD’s biological basis, therapeutic relevance and dual role both protective and pathological.
  2. The abstract mentions diagnostic biomarker potential and personalized medicine but lacks specificity. Briefly specify how NMD modulation such as small molecule inhibitors or antisense oligonucleotides might aid gene therapy.
  3. Line 149 “downregulates its substrate by 2-20 folds”. Specify context or give example genes.
  4. Lines 171-173 “The diverse and context-dependent roles of NMD in development warrant specific attention, as individual NMD factors exert distinct effects that cannot be generalized.” Support with examples or citations.
  5. Authors should relate the NMD-apoptosis link to disease states (e.g., cancer, ischemia) for better translational framing.
  6. Line 206 “Simultaneous suppression of NMD and stimulation of autophagy initiates apoptosis and performs a protective role”. Further clarify. The sentence seems confusingly dual.
  7. Lines 234-236. “However, inhibiting or bypassing NMD restores the expression of deleterious PTC-mutant proteins, such as HSP110DE9, thereby suppressing MSI tumor growth”. Clarify if restoring expression of mutant proteins is beneficial or harmful, since it depends on context.
  8. Explain the link between altered NMD and its impact on diagnostic/prognostic strategies with example.
  9. The title “NMD Escape by Immune Cells” is slightly misleading. Immune cells don’t escape NMD, rather NMD escape promotes immune recognition. Revise the title to: “NMD Escape and Its Role in Antitumor Immunity”
  10. Line 524 “Currently been investigated" should be “currently under investigation”.
  11. How does FMRP loss typically studied in fragile X syndrome translate mechanistically into cardiac implications? Explain briefly
  12. Line 599 "approach that worth being considered" should be "approach worth considering" or "that is worth considering".
  13. Line 606 determine the exact point which NMD should be inhibited..." revise to "determine the threshold at which NMD should be inhibited..
  14. Line 611 This could be an approach to prevent diseases caused by the presence of aberrant proteins that escaped NMD..." — this is unclear. Be more specific about disease examples and mechanisms.
  15. The title “Nonsense-Mediated mRNA Decay: Mechanisms and Recent Implications in Cardiovascular Diseases” does not reflect the broader disease spectrum covered in the manuscript. If CVD is still the primary focus but others are discussed secondarily, change the title to "Nonsense-Mediated mRNA Decay: Mechanistic Insights and Emerging Implications in Cardiovascular and Other Human Diseases".

Author Response

We thank the reviewer for positive feedback and the suggestions. In the revised version, we have condensed the content and improved the sentence structure to effectively convey the central role of NMD in cardiovascular diseases and emphasize its significance as a therapeutic target. We have thoroughly considered each of your suggestions and made the necessary revisions to address the raised issues. 

Comment 1: In abstract, authors should slightly expand the abstract to clearly explain NMD’s biological basis, therapeutic relevance and dual role both protective and pathological.

Response: Biological basis of NMD has been explained and The dual effect of NMD has been stated in the abstract.

Comment 2: The abstract mentions diagnostic biomarker potential and personalized medicine but lacks specificity. Briefly specify how NMD modulation such as small molecule inhibitors or antisense oligonucleotides might aid gene therapy.

Response: This has been addressed in the review.

Comment 3:  Line 149 “downregulates its substrate by 2-20 folds”. Specify context or give example genes.

Response: This statement has been clarified with citation added.

Comment 4: Lines 171-173 “The diverse and context-dependent roles of NMD in development warrant specific attention, as individual NMD factors exert distinct effects that cannot be generalized.” Support with examples or citations.

Response: Citation added

Comment 5 Authors should relate the NMD-apoptosis link to disease states (e.g., cancer, ischemia) for better translational framing.

Response 5: The segment has been linked to a disease state and condition with reference provided.

Comment 6: Line 206 “Simultaneous suppression of NMD and stimulation of autophagy initiates apoptosis and performs a protective role”. Further clarify. The sentence seems confusingly dual.

Response: It is a confusing statement and has been removed.

Comment 7:Lines 234-236. “However, inhibiting or bypassing NMD restores the expression of deleterious PTC-mutant proteins, such as HSP110DE9, thereby suppressing MSI tumor growth”. Clarify if restoring expression of mutant proteins is beneficial or harmful, since it depends on context.

Response: This statement has been clarified, as restoring this mutant protein has been reported to play a beneficial role in suppressing MSI CRC tumor growth.

Comment 8: Explain the link between altered NMD and its impact on diagnostic/prognostic strategies with example.

Response: The possibility of improved treatment has been explained by example given with MSI CRC cancer.

Comment 9:The title “NMD Escape by Immune Cells” is slightly misleading. Immune cells don’t escape NMD, rather NMD escape promotes immune recognition. Revise the title to: “NMD Escape and Its Role in Antitumor Immunity”

Response: This correction has been effected.

Comment 10:Line 524 “Currently been investigated" should be “currently under investigation”.

Response: Done

Comment 11 : How does FMRP loss typically studied in fragile X syndrome translate mechanistically into cardiac implications? Explain briefly

Response: FMRP loss has been reported to disrupt NMD in neurodegenrative diseases, however it could be explored if similar proteins could affect cardiovascular diseases due its effect on major NMD factors.

Comment 12 :Line 599 "approach that worth being considered" should be "approach worth considering" or "that is worth considering".

Response: Corrected.

Comment 13: Line 606 determine the exact point which NMD should be inhibited..." revise to "determine the threshold at which NMD should be inhibited..

Response: Changes made

Comment 14: Line 611 This could be an approach to prevent diseases caused by the presence of aberrant proteins that escaped NMD..." — this is unclear. Be more specific about disease examples and mechanisms.

Respone: An example has been added with citation.

Comment 15:  The title “Nonsense-Mediated mRNA Decay: Mechanisms and Recent Implications in Cardiovascular Diseases” does not reflect the broader disease spectrum covered in the manuscript. If CVD is still the primary focus but others are discussed secondarily, change the title to "Nonsense-Mediated mRNA Decay: Mechanistic Insights and Emerging Implications in Cardiovascular and Other Human Diseases".

Response: The title and review have been modified to address this concern.

Reviewer 3 Report

Comments and Suggestions for Authors

This is a well-written review of NMD and its effect on human diseases.  The focus on cardiovascular disease seemed somewhat inappropriate since NMD is also involved in all other organs.  And it has both positive and negative effects, so attempts to inhibit it would likely be harmful, even if it could be targeted to the heart.

  1.  line 413: redundant with line 414
  2. line 437:  What SARS protein?

Author Response

We tremendously thank the reviewer for the positive feedback. Necessary changes have been made.

Comment 1: line 413: redundant with line 414

Response: This has been addressed

Comment 2: line 437:  What SARS protein?

Response: This segment of Viral infection has been modified and corrected.

Round 2

Reviewer 1 Report

Comments and Suggestions for Authors

The manuscript has improved following the previous revision; however, it still requires substantial editing to address issues related to scientific accuracy, appropriate use of terminology, and overall writing quality. One example of poor academic writing is the repeated and inconsistent use of terminology—abbreviations such as “nonsense-mediated mRNA decay (NMD)” and “premature termination codon (PTC)” are redefined multiple times after their initial introduction, which is unnecessary and distracting.

Given the combination of scientific inaccuracies, lack of consistency in writing, and serious concerns about originality, I cannot recommend this manuscript for publication in its current form.

Below, I provide several representative examples to illustrate the types of issues present. Please note that this list is not exhaustive.

________________________________________

Representative Examples:

Line 14 (Abstract):

The authors should include both major models of NMD activation, not just the EJC-dependent model. A more accurate and inclusive statement would be:

"NMD is a highly conserved surveillance mechanism that degrades mRNAs containing premature termination codons (PTCs) located upstream of the final exon-exon junction or at a long distance from the poly(A) tail."

Line 52:

The term “premature termination codon” appears for the first time and should be written as “premature termination codon (PTC)” here, not later at lines 54–55.

Line 56:

“EJCs” should be revised to “last exon-exon junction.”

The authors appear to conflate spatial distance from exon-exon junctions with the physical presence of exon junction complexes (EJCs). These concepts are not interchangeable.

Line 58:

“Final exon junction” should be corrected to “final exon.” An exon junction refers to the boundary between two exons, not an individual exon.

Line 59:

“Nonsense-mediated mRNA decay (NMD)” should be abbreviated to “NMD” here, since the term has already been defined.

Line 62:

This section contains incorrect mechanistic details. Specifically, UPF2 and UPF3 interact with the NMD machinery prior to UPF1 phosphorylation, whereas SMG5–7 bind after UPF1 has been phosphorylated. Additionally, the interactions between UPF proteins and the EJC are not described correctly. The sequence and timing of these events should be revised for accuracy.

Lines 78 and 84:

Replace “premature stop codon (UAA)” with “premature termination codon (PTC).” Using a specific stop codon is misleading in this context.

Line 80:

The acronym “SURF” should be defined upon first use—for example:

“SURF (SMG1–UPF1–eRF1–eRF3 complex).”

Line 86:

The manuscript inconsistently uses “poly A” and “poly(A).” The standard usage is “poly(A)” and should be applied consistently throughout.

Line 90 and Figure 1:

Using “UAA” as a representative stop codon is misleading. To maintain accuracy and generality, the figure and text should refer to “PTC.”

Line 97:

Replace “the Poly A binding protein” with “the poly(A)-binding protein (PABPC1).”

Line 98:

Since “eukaryotic release factor 3 (eRF3)” was already defined earlier, simply use “eRF3” here.

Line 124:

This section focuses on examples of genetic disorders related to NMD, which does not align with the section title “NMD and Regulation of Gene Expression.” The title or content should be revised to better match the section’s focus.

Author Response

We thank the reviewer for their careful evaluation of our manuscript and for the constructive comments and suggestions provided. Please find our point-by-point responses below.

Line 14 (Abstract):

The authors should include both major models of NMD activation, not just the EJC-dependent model. A more accurate and inclusive statement would be:

"NMD is a highly conserved surveillance mechanism that degrades mRNAs containing premature termination codons (PTCs) located upstream of the final exon-exon junction or at a long distance from the poly(A) tail."

Response: We thank the reviewer for this important suggestion. The abstract has been revised to include both major models of NMD activation, as recommended.

Line 52:

The term “premature termination codon” appears for the first time and should be written as “premature termination codon (PTC)” here, not later at lines 54–55.

Response: We appreciate the reviewer’s attention to terminology. “Premature termination codon (PTC)” is now defined at its first appearance, as suggested. It was referred to as stop codon initially (Line 13) and PTC is defined in line 54.

Line 56:

“EJCs” should be revised to “last exon-exon junction.”

The authors appear to conflate spatial distance from exon-exon junctions with the physical presence of exon junction complexes (EJCs). These concepts are not interchangeable.

Response: This has been corrected as advised.

Line 58:

“Final exon junction” should be corrected to “final exon.” An exon junction refers to the boundary between two exons, not an individual exon.

Response: Correction has been made as suggested.

Line 59:

“Nonsense-mediated mRNA decay (NMD)” should be abbreviated to “NMD” here, since the term has already been defined.

Line 62: (Comment 6)

This section contains incorrect mechanistic details. Specifically, UPF2 and UPF3 interact with the NMD machinery prior to UPF1 phosphorylation, whereas SMG5–7 bind after UPF1 has been phosphorylated. Additionally, the interactions between UPF proteins and the EJC are not described correctly. The sequence and timing of these events should be revised for accuracy.

Response: Thank you for highlighting this issue. The mechanistic details in this section have been revised for accuracy, and the figure has been updated accordingly

Lines 78 and 84:

Replace “premature stop codon (UAA)” with “premature termination codon (PTC).” Using a specific stop codon is misleading in this context.

Response: Correction made as suggested

Line 80:

The acronym “SURF” should be defined upon first use—for example:

“SURF (SMG1–UPF1–eRF1–eRF3 complex).”

Response: Correction made as suggested

Line 86:

The manuscript inconsistently uses “poly A” and “poly(A).” The standard usage is “poly(A)” and should be applied consistently throughout.

Response: All instances have been standardized to “poly(A)” throughout the text

Line 90 and Figure 1: (Comment 10)

Using “UAA” as a representative stop codon is misleading. To maintain accuracy and generality, the figure and text should refer to “PTC.”

Response: Noted and corrected in both the figure and text

Line 97:

Replace “the Poly A binding protein” with “the poly(A)-binding protein (PABPC1).”

Response : Correction made as suggested.                                                                                        

Line 98: (Comment 12)

Since “eukaryotic release factor 3 (eRF3)” was already defined earlier, simply use “eRF3” here.

Response: The text now uses “eRF3” in all subsequent references.

Line 124: (Comment 13)

This section focuses on examples of genetic disorders related to NMD, which does not align with the section title “NMD and Regulation of Gene Expression.” The title or content should be revised to better match the section’s focus.

Response: The section content has been revised to better reflect and align with its title.

Reviewer 3 Report

Comments and Suggestions for Authors

Acceptable with modifications

Author Response

We thank the reviewer for the continuous support. Modifications were addressed as requested.